# Evaluation of Traits for the Selection of Apis Mellifera for Resistance against Varroa Destructor

**DOI:** 10.3390/insects11090618

**Published:** 2020-09-10

**Authors:** Ralph Büchler, Marin Kovačić, Martin Buchegger, Zlatko Puškadija, Andreas Hoppe, Evert W. Brascamp

**Affiliations:** 1Landesbetrieb Landwirtschaft Hessen, Bee Institute, Erlenstrasse 9, 35274 Kirchhain, Germany; 2Faculty of Agrobiotechnical Sciences Osijek, University of Osijek, Vladimira Preloga 1, 31000 Osijek, Croatia; Marin.Kovacic@fazos.hr (M.K.); zpuskadi@fazos.hr (Z.P.); 3Department of Sustainable Agricultural Systems, University of Natural Resources and Life Sciences Vienna (BOKU), Division of Livestock Sciences, Gregor-Mendel-Straße 33, 1180 Vienna, Austria; buchegger29@gmail.com; 4Institute for Bee Research, Friedrich-Engels-Str. 32, 16540 Hohen Neuendorf, Germany; andreas.hoppe@hu-berlin.de; 5Animal Breeding and Genomics, Wageningen University & Research, P.O. Box 338, 6700 AH Wageningen, The Netherlands; pim.brascamp@wur.nl

**Keywords:** Varroa resistance, mite infestation, brood infestation, pin test, suppressed mite reproduction (SMR), brood recapping, repeatability, performance test, breeding objective

## Abstract

**Simple Summary:**

Infestation with the parasitic mite *Varroa destructor* is a serious cause of bee colony (*Apis mellifera*) losses on a global level. However, the presence of untreated survivor populations in many different regions indicates that selection for resistance might lead to a long-term solution. The success of selection depends on suitable testing criteria. To be effective, results must show repeatable effects of the individual genotype and correlate with the breeding goal. As colony survival is difficult to measure, selective breeding for Varroa resistance can be based on differences in mite infestation and specific behavioral traits. In this paper we look into different definitions of mite infestation and link these with brood hygiene (pin test), brood recapping and suppressed mite reproduction (SMR). Due to the large dataset (489 colonies) from Austria, Croatia and Germany and four seasons (2016–2019), our study arrogates high representability. Repeatability analysis depicts different infestation parameters, brood hygiene and recapping data as characteristic colony traits while SMR results are very strongly influenced by environmental effects. Brood hygiene and recapping data correlate weakly but significantly with mite infestation. We therefore recommend combining them with estimates of mite population increase and brood infestation for an effective selection on resistance.

**Abstract:**

Infestation with *Varroa destructor* is a serious cause of bee colony (*Apis mellifera*) losses on a global level. However, the presence of untreated survivor populations in many different regions supports the idea that selection for resistance can be successful. As colony survival is difficult or impossible to measure, differences in mite infestation levels and tests for specific behavioral traits are used for selective breeding for Varroa resistance. In this paper we looked into different definitions of mite infestation and linked these with brood hygiene (pin test), brood recapping and suppressed mite reproduction. We based our analyses on datasets of *Apis mellifera carnica* from three countries: Austria (147 records), Croatia (135) and Germany (207). We concluded that bee infestation in summer, adjusted for the level of natural mite fall in spring, is a suitable trait in the breeding objective, and also suggested including brood infestation rate and the increase rate of bee infestation in summer. Repeatability for bee infestation rate was about 0.55, for cells opened in pin test about 0.33, for recapping 0.35 and for suppressed mite reproduction (SMR) virtually zero. Although in most cases we observed correlations with the expected sign between infestation parameters and behavioral traits, the values were generally low (<0.2) and often not significantly different from zero.

## 1. Introduction

Resistance against *Varroa destructor* of its natural host, *Apis cerana* [1], and of numerous local populations of its new host, *Apis mellifera* [2], suggests that this worldwide threat for bee keeping can be overcome by selective breeding. However, several challenges complicate this strategy. Firstly, the interactions between mites and bees are complex, including environmental effects such as climate, food resources and nesting sites, which lead to local adaptation and genotype by environment interaction. Consequently, a simple transfer of queens or colonies from a resistant population to other regions or living conditions usually fails to maintain a high level of resistance [3]. Secondly, in modern bee keeping, many practices, as for example swarm prevention, support of continuous brood activity, high colony density and large colony size, are hampering the establishment of mite resistance. Thirdly, there is the issue of the target trait or sets of traits to select for—the breeding objective. The ultimate objective of selection for Varroa resistance, the golden standard, is a high survival of colonies without curative treatments. Under natural conditions, survival in a given setting of environmental conditions serves as a direct selection trait. However, most beekeepers are not willing to accept a high risk of losing colonies, and in many countries, the regular use of therapeutic measures against varroosis is prescribed by veterinary legislation.

Consequently, the golden standard is a trait that cannot be measured under practical circumstances and alternative traits are needed as part of an operational breeding objective. Since the early 1990s, a number of potential resistance mechanisms were described, such as hygienic behavior, reduced postcapping stage duration of worker brood, grooming behavior, Varroa sensitive brood hygiene, recapping of brood (REC) and suppressed mite reproduction (for more details see reviews [4,5]). High expression of these traits in a bee colony can either reduce the life expectancy or the reproductive success of mites, and consequently, can be expected to result in reduced mite population growth in the colony over time. For all these traits, effects on mite population development and a certain degree of repeatability have been shown in smaller studies. However, most of these traits are not easy to measure under field conditions, and therefore, larger datasets involving variable environmental conditions and genetic stock are not yet available.

In order to study relationships between these traits and mite infestation we evaluated performance test data from three independent Carnica breeding populations in Austria, Croatia and Germany, collected during 2016–2019. Traits measured included natural mite mortality (NMF) in early spring, repeated bee infestation (BINF) measured during summer and repeated pin test on brood hygiene behavior (PIN). Furthermore, one mite infested brood comb of each colony was evaluated for mite reproduction (SMR), recapping of all investigated cells (RECall), and recapping of mite infested cells (RECinf). In the Croatian dataset REC and SMR were measured repeatedly. In total, the dataset comprised 489 colonies.

In this paper, we took mite infestation as an element in the operational breeding objective in line with the extensive literature review of Guichard et al. [6], who emphasized that a proper definition of resistance for selection purposes is not without problems. We compared different definitions of mite infestation and analyzed to what extend PIN, REC and SMR could explain variation in mite infestation. We started this study with the analysis of repeatability of traits, because the degree to which repeated measurements of traits resemble each other has consequences for their usefulness to evaluate colonies and also for the level of correlations with other traits we might expect.

We discussed the consequences of our findings for selection traits and the design of performance testing.

## 2. Materials and Methods

### 2.1. Genetic Set-Up and Colony Management

Austria—Queens tested in Austria originated from highly selected Austrian Carnica stock participating in the AGT-breeding program (www.toleranzzucht.de). Queens were either artificially inseminated or open mated at isolated mating stations. Queens were registered in the BeeBreed database (www.beebreed.eu). In the three seasons, 2017–2019, a total of 147 colonies were subjected to performance testing for one year each in 18 apiaries of five different breeders. Test queens were introduced into the colonies in July one season before testing, either by replacing the existing queen or by creating a new colony as shook swarms with 2 kg of bees. Treatment against *V. destructor* was performed with formic or oxalic acid in summer and with oxalic acid in winter. Each colony was able to rear drones on one frame with drone comb, which was not manipulated.

Croatia—Queens tested in Croatia derived from one registered Carnica breeder located near Osijek in northeast Croatia. All queens were open mated at a geographically non-isolated mating station that was saturated with drones of known origin. Queens were not selected for SMR or REC prior to 2016 and were selected for hygienic behavior as measured by the pin test for only one generation. Testing was performed during four seasons (2016–2019) with a total of 135 colonies located in three apiaries. Colonies were formed in May in the season before performance testing. During that summer, colonies were treated against *V. destructor* with registered drugs (CheckMite^®^ or Bayvarol^®^). During the broodless period in winter, colonies were treated by trickling oxalic acid. After the winter treatment, no further treatment against *V. destructor* was performed until August of the test season. In 2016, six repeated brood samplings from April until September were made.

Germany—Queens tested in Germany originated from the Bee Institute in Kirchhain and several registered breeders who all were members of the regional AGT breeding group. All queens derived from pure Carnica stock with long time pedigrees registered in the BeeBreed database and were mated with selected Carnica drones either by artificial insemination or at island mating stations. A total of 207 colonies was subjected to performance testing in three apiaries during four seasons (2016–2019). All colonies were started as artificial swarms with 2 kg of bees in early July of the season before performance testing. A treatment with Coumaphos was applied on the swarms to start the testing with a low Varroa infestation level. However, no further treatments against Varroa were carried out until the end of July of the following season. At this time, all queens were caged and oxalic acid was trickled onto the broodless colonies about 25 days later.

### 2.2. Performance Testing

To enable an overview, all relevant traits and their abbreviations are listed in Table 1. Furthermore, Figure 1 illustrates the involvement of traits in the data analysis procedure.

### 2.3. V. destructor Infestation Level

The infestation level of colonies with *V. destructor* mites was determined by three different measurements: natural mite fall (NMF) in spring, worker bee infestation (BINF) in summer and worker brood infestation (BRINF).

NMF was monitored using sticky sheets on bottom boards during the blossom of *Salix caprea* in three consecutive weeks [7].

The infestation rate of adult workers in Austria was determined by the powder sugar shake method [8], and in Croatia and Germany by the soapy water wash method [9]. In Austria, BINF measurements (BINF1–5) started in early July with repeated measurements every three weeks. Up to five measurements were performed unless an infestation threshold was reached earlier (guideline: no threshold at the begin of July/BINF1; 1.5 mites per 10 g bees at the end of July/BINF2; 2.5 mites per 10 g bees at the mid of August/BINF3; 3.4 mites per 10 g bees at the begin of September/BINF4). In Croatia, three measurements (BINF1, BINF2 and BINF3) were performed in 2016 and only BINF1 in 2017 and 2018. In Germany, three measurements (BINF1, BINF2 and BINF3) were performed in three-week intervals, starting in early June. The second measurement was combined with caging the queen. Consequently, in Germany BINF3 was recorded while colonies were mostly free of brood. Despite the apparent differences in BINF-measurements between countries we consider them as representing the same trait. The protocol in principle is the same in which for example the powder sugar shake method and the soapy water wash method are considered to measure the same trait and the differences in timing are a function of the climatic differences between the countries. In the German dataset, BINF3 was measured after caging the queen and as a consequence average BINF3 was disproportionately bigger than average BINF1 and BINF2. When combining BIN1, BINF2 and BINF3, we adjusted BINF3 with a factor 4.31, which is the ratio between average BINF3/BINF2 and BINF2/BINF1. In that way, in average, the adjusted values follow an exponential growth path.

Brood infestation was measured on samples of brood collected for the investigation of REC and SMR. On average, 264 cells were individually opened and checked for the presence of any adult female mite. The rate of infested cells from all opened cells was registered as brood infestation (BRINF).

In Germany, the mites killed within 14 days after oxalic acid trickling (MFOA) of the broodless colonies were collected with sheets placed under the wired bottom boards. Its total number was used to estimate the colony infestation in addition to the preceding BINF measures.

### 2.4. Pin Test

Hygienic behavior was measured using a standard pin test method [10,11]. During the first three seasons, the timing of inspection of the pierced cells was set to an interval at which the apiary average of emptied cells was expected to reach about 50%; this was the case after eight hours in Austria and Germany and after 16–18 h in Croatia. In 2019, all colonies were checked six hours after piercing. The rates of opened (PINop) and of completely emptied cells (PINem) were determined. In Austria two pin tests were usually performed per season (one in June and one in July). In Croatia one is usually performed in June (except 2016, when three tests were done, in May, June and July) and in Germany two (one in early May and one mid-June).

### 2.5. Suppressed Mite Reproduction and Recapping

Suppressed mite reproduction (SMR) and recapping of brood cells (REC) were measured according to the RNSBB-protocol [12]. The brood samples in Austria were collected in August and September, in Croatia in July and August and in Germany in July. In 2016 in Croatia, six repeated brood samplings were collected from mid-May until September with three-week intervals. Brood cells at a minimum development stage of 7 days post capping were examined under a stereo microscope (magnification 5×–10×). The cap of the brood cell was carefully removed with fine forceps and it was noted if part of the pupa’s cocoon was lacking on the inner side of the cell capping. This was used as an indication that a brood cell had been manipulated (recapped) by bees during the development of the pupa [13,14]. This gave rise to RECall, recapping as a fraction of all inspected cells. Furthermore, pupae were removed from the cell and checked for infestation with *V. destructor*. This recording was used to calculate RECinf, recapping as a fraction of infested cells. Only brood cells infested with a single foundress mite were considered in the current SMR analysis. When there was a mite in the brood cell, development stages of the pupae and *V. destructor* offspring were noted. A foundress mite was considered non-reproductive if: (1) there was no offspring or exclusively male offspring (infertility), (2) offspring was too young to mature until eclosure of the bee (delayed development) or (3) the male was absent. The data were also used to estimate the brood infestation (BRINF) as described above. Sometimes, the brood sample was too small or the infestation was too low to detect sufficient single infested cells for SMR analysis. In total, we got 486 samples with at least 10 single infested cells, and out of them 381 samples with at least 25 single infested cells. The average sample size was 23.9 single infested cells.

The numbers of observations, means and standard deviations are summarized in Table 2.

Comparing the standard deviations with the means lead to the conclusion that the distributions of the traits were not symmetric. For the correlation analysis we transformed all data according to the Box-Cox algorithm, but as this affected correlations only slightly, we decided to analyze the data in units of measurement. The means for NMF and BINF were higher in the German dataset than in the other two as a consequence of no winter treatment against Varroa in Germany in the season prior to testing. Surprisingly, this was not reflected in the mean of BRINF, which was very similar in the German and Austrian datasets but lower in the Croatian one. Means for PINem and PINop seemed fairly similar across countries, probably as a consequence of selecting a time interval for checking that aimed at a PINem of about 50%. The SMR values also did not differ much between countries; however, they were quite low as it is characteristic for mainly populations not selected for SMR.

### 2.6. Preparation of the Data Preceding Correlation Analyses

Repeatability, properties of different definitions of mite infestation and the degree to which variation in mite infestation could be explained by REC, SMR and PIN were studied by looking at deviations. To arrive at deviations all traits were adjusted for the effects of season and apiary by subtracting an appropriate mean from each observed value as explained below. All analyses were carried out for the three countries separately. Although the recording methods were standardized, the colonies in each country were of different origins, and the environmental situations differed. Comparison of results between countries might provide clues to what degree phenomena are general or specific.

The data were modified in three steps to arrive at deviations.

In the first step three additional traits were defined. These were the mite-population growth rate (MPG) and two growth parameters, b3 and b5.

The mite-population growth rate was computed separately for each of the three mite infestation scorings BINF1, BINF2 and BINF3. For example, for BINF1:(1)$$\mathrm{MPGi}=\ln\left(1+10\frac{1+\mathrm{BINFi}}{1+\mathrm{NMF}}\right)$$

Subsequently, the three MPGi-values were averaged to arrive at the overall MPG. Please note that for the German dataset BINF3 was divided by 4.31 before computing MPG3 because BINF3 was measured when hardly any or no brood at all was present.

The growth parameters b3 and b5 described the exponential growth of mite infestation as:(2)$$\mathrm{BINFi}={\mathrm{ae}}^{\mathrm{bt}}$$
where the parameter a described the overall level of the exponential growth, t is time (which we take to be 1, 2, …, 5) and b is the growth parameter. For all three datasets, b3 was estimated based on BINF1, BINF2 and BINF3. In addition, in the Austrian dataset, b5 was estimated based on all five BINF-measurements. We estimated b from the linear regression:(3)ln(BINFi)=a+bt+residual

Overall b3 and b5 were computed as the average of the b’s considering BINF1–3 or BINF1–5, respectively.

In a second step we analyzed the data to look at the statistical significance of the effects of season and apiary, starting with the model:(4)$$y_{\mathrm{ijk}}=\mu +{\mathrm{season}}_{i}+{\mathrm{apiary}}_{j}+{\left(\mathrm{season}\;x\;\mathrm{apiary}\right)}_{\mathrm{ij}}+e_{\mathrm{ijk}}$$

Deviations were calculated as the differences between the observations and their appropriate means. In case the interaction term was statistically significant (*p* < 10%), observations were adjusted for the season x apiary subclass mean. If that was not significant, but season alone was, adjustment was made for the applicable season mean, and if apiary alone was significant, for the apiary mean. If season and apiary were significant adjustment was made for the appropriate least square mean from the statistical model only including season and apiary. For details of the relevant adjustment see Appendix A.

In Croatia a number of traits was only observed in one apiary in one year, such that adjustment was only for the overall mean.

The result of the steps 1 and 2 was a dataset with adjusted traits that were subject of further analysis. To this file, a last trait was added, BINFa, which is BINF adjusted for NMF. For each of BINF1, BINF2 and BINF3 within each country the regression coefficient of BINFi on NMF was computed. Subsequently, the respective BINF was adjusted according to Equation (5), as:(5)$${\mathrm{BINFi}}_{\mathrm{adjusted}}=\mathrm{BINFi}-b_{\mathrm{BINFi}.\mathrm{NMF}}\mathrm{NMF}$$

BINFa was computed for each colony as the average of the three values for ${\mathrm{BINFi}}_{\mathrm{adjusted}}$. For the German dataset, as before, BINF3 was divided by 4.31.

The adjusted traits were analyzed calculating Pearson correlations between pairs of traits. Standard errors (se)  of these correlations were calculated as:(6)$$\mathrm{se}=\frac{1-r^{2}}{\sqrt{N-2}}$$
where r is the Pearson correlation between two traits and N is the number of cases where both traits were recorded. For a table with correlations between all pairs of traits with standard errors see Appendix A. This supplementary table also contains the 90% and 99% confidence intervals.

## 3. Results

In this paragraph, we stepwise present the results, ultimately arriving at the focal objective of our paper, examining the correlations between behavioral traits and mite infestation. To be able to do this, we firstly present the repeatability of traits concerned, analyzed different definitions of mite infestation and looked at correlations between behavioral traits. This setup is illustrated in Figure 1.

### 3.1. Repeatabilities

To estimate the robustness and reliability of measurements that are performed multiple times, the repeatability was calculated as the correlation between different measurements of the same trait. This was calculated individually, between each pair of observations to distinguish between different time points and exhibit a dependence on time distance. The qualitative consensus of individual correlations was preferred to the overall correlation for a more complete picture of the situation.

See Table 3 for these correlations for BINF (data for BINF 4–5 are included in Appendix A). In the dataset from Germany, the correlations were in the order of magnitude of 0.55, but in the Croatian data, they were significantly higher. Correlations between subsequent measurements tended to be somewhat higher compared to measurements that were taken further apart, but the differences were small. However, in the Austrian data this phenomenon was clearly present. Here, the numbers of observations were considerably lower for BINF4 and BINF5 than for the earlier three readings, and for this reason BINF1–BINF3 were used to define mite-infestation traits. Clearly, for b5, all five readings were included

The correlations between different readings for PINem and PINop were smaller than those for BINF, in the range of 0.25 (Table 4).

The correlation between the second and third Croatian PINem (data in Appendix A) was lower (0.09) than between the other two, but not significantly so, as the standard error was 0.15. The repeatability of PINop seemed somewhat larger than the one of PINem, although not significantly, considering the standard errors.

Correlations for repeated measurements of RECall and RECinf were only calculated for the Croatian dataset (Table 5).

We observed a tendency towards weaker correlations between RECall measurements when the readings were taken further apart. For RECinf, that seemed not to be the case, but both the standard errors were large. Overall repeatability had an order of magnitude of 0.35.

The correlations between repeated measurements of SMR also could only be calculated for the Croatian dataset (Table 6). Just as in the case of REC, standard errors were high. It was difficult to discover consistency in the different correlations, but it seemed fair to conclude that the repeatability was very low.

### 3.2. Correlations between Infestation Traits

The correlations between different infestation traits were calculated to study the degree of similarity between these traits and to lay the foundation to understand the relations of behavioral traits with infestation traits. Due to the differences in number and type of measurements, the definition of infestation traits differed between the countries. The trait b5 is calculated on five BINF measurements only in Austria. Mite fall after oxalic acid treatment MFOA was only assessed in Germany. For Germany, BINF3 was also included in this analysis as it measured the full infestation due to the absence of brood.

See Table 7 for all correlations between the infestation traits. The correlations between BRINF and the other traits mostly had a positive sign, as might be expected. In the German dataset, BRINF correlated fairly high with MPG and BINFa, but this was not the case in the other two datasets. Moreover, the additional parameters BINF3 and MFOA are highly correlated with BRINF in the German dataset.

The correlation between MPG and BINFa was quite high for all three datasets. Both for MPG and BINFa, the adjustment for NMF intended to make mite population growth independent of initial mite infestation. Clearly, for MPG, this did not work as intended, but for BINFa it did, as expected when adjusting by linear regression. B3 on the one hand and MPG and BINFa on the other hand had low correlations in the German dataset, but higher ones in the Austrian and Croatian datasets. In all three datasets the correlation differed very significantly from 1, suggesting that mite-population growth from spring to summer (as measured by NMF in spring and BINF in summer), and during summer (as estimated by repeated readings of BINF), are two different traits. In the German dataset the correlation between MFOA and BINF3 (both counting total mite numbers, although with different methods) was 0.36, apparently expressing the total number of mites either proportional or not to colony size affected the trait considerably.

### 3.3. Correlations between Behavioral Traits

See Table 8 for all correlations between the behavioral traits. The correlations between PINem and PINop were about 0.45 in the three datasets. The correlations of the two PIN-traits with other traits partly differed between the countries. In the Austrian dataset the correlation between PINem and SMR was significantly lower than the one between PINop and SMR. That was not the case in Croatia and Germany, however. The correlations between RECall and RECinf were about 0.80 in all three countries. The correlations between the REC-traits and SMR differed considerably between countries, however. In the Austrian dataset the correlation was about 0.40, but in the other two countries it was not significantly different from zero. In general, RECinf showed higher correlation with PIN traits and SMR than RECall.

### 3.4. Relationships between Behavioral Parameters and Mite Infestation

In this paragraph we limit ourselves to the traits that were available in the datasets from all three countries: BRINF, BINFa and b3. BINFa was included because this was clearly independent from NMF, but MPG was not. The objective of this paragraph was the correlation between the mite-infestation traits BRINF, BINFa and b3 and the behavioral traits PINem, PINop, RECall, RECinf and SMR. Results are presented in Table 9.

Overall, the correlations between BRINF and the behavioral traits mostly had a negative sign, as expected. Exceptions are the correlations with RECall in Croatia and Germany that were, however, not significantly different from zero. The correlations between the mite infestation traits with PINem and PINop also had the expected sign, with the exception of b3 in Germany that was not significantly different from zero. Generally speaking, the correlations of REC and SMR with BINFa and b3 were low and often not significantly different from zero.

## 4. Discussion

Very generally, large effects of the environment will lower the repeatability and inhibit high correlations. Behavior and expression of resistance traits, but also varroa reproduction success [15,16] and drifting of mite infested bees between colonies [17,18], are strongly influenced by environmental factors. Data analyzed in this study was collected at 24 apiaries in three countries over 4 years, causing large environmental differences. The differences between countries, years and apiaries were eliminated as described in Table 2, but the remaining environmental variation is still considerable and should be taken into account when interpreting low correlations.

### 4.1. Repeatability

The measurements of mite infestation on bees and of the behavioral traits were repeated, such that the repeatability could be studied. As the behavioral traits can be assumed to represent a constant genetic disposition, information on the impact of the environmental conditions and the technical robustness of the measurement can be learned. Mite measurements, however, monitor a dynamic system; thus, in addition to factors influencing the measurements itself, divergent development routes decrease the correlation between repeated readings, and subsequent measurements can be expected to correlate more with each other than those lying further apart. Arguably, mite infestations might also affect behavioral traits, which influence the repeatability of their measurements.

The mite infestation measurements BINF1-3 were highly repeatable, with correlation ranging from over 0.50 to 0.85. This is in general agreement with the high repeatability of BINF (0.85) reported by Büchler et al. [4], although slightly lower. Regarding the repeatability of mite infestation measurements for breeding colonies registered in BeeBreed with at least three mite infestation measurements, the correlation of BINF1–BINF2 was 0.58 ± 0.01, of BINF1–BINF3 0.47 ± 0.01 and of BINF2–BINF3 0.45 ± 0.01 (Hoppe, unpublished data 2020). Thus, the repeatability of BINF in BeeBreed data is similar to that in our study, although the standards of data acquisition were lower in BeeBreed, e.g., the breeders can freely choose the frequency and time points of measurements.

Correlations between mite infestation readings late in the season (BINF4 and BINF5 in Austria) were considerably lower with factors from 0.1 to 0.2. This can be interpreted in a way that factors controlling the mite reproduction in summer, the time of strongest brood activity, are different from factors affecting mite infestation later in the year in a time of lesser brood activity. An important factor is seen in an increasing risk of mite transmission between colonies in the course of season [19]. However, it has to be noted that this interpretation is taken with caution because it is based on few colonies.

As expected, the subsequent BINF measurements are more highly correlated than nonadjacent measurements, with few exceptions. One exception is BINF4 in the Austrian data, where again the small number of observations is to be considered. The other exception is BINF3 in Croatia, where it has to be noted that the correlations are much higher than for the other countries, and the variance of the measurement itself may be more relevant.

For the repeatability of measurement of brood hygiene with the pin test, we found a large difference depending on which cells were counted. Counting only fully cleared cells (PINem), the correlation was between 0.1 and 0.3, while counting all cells that were at least opened (PINop), the correlations were considerably higher (between 0.3 and 0.4). We interpret this as a higher reliability of PINop, and those findings have prompted the German breeder association “Arbeitsgemeinschaft Toleranzzucht—AGT” to replace PINem with PINop for its test protocol [20]. The repeatability for different methods were also compared by Hoffmann [21]. PINem had the highest repeatability, with 0.54, while the repeatability for cells with artificial mite infestations was 0.38, and the repeatability for freeze-killed brood just 0.10. In a continuation study, a similar repeatability of 0.55 was found in a genetically diverse Carnica population of 69 colonies [21]. Boecking et al. [22] reported a repeatability of 0.46.

Interestingly, the repeatability of PINem decreased over time, which can be interpreted as the effect of selection for hygienic behavior, where the time needed to remove 50% of the treated cells decreased on average from about 16 h in 1994 to about 8 h in recent years. While in 1994 the repeatability was reported as 0.54 [18] for colonies at the institute in Kirchhain (part of the AGT-program), in the continuation program it was 0.28 (Büchler, unpublished data 2009), and here it was only 0.20.

The repeatability of recapping rates, measured in Croatia, ranged from 0.06 to 0.95 and was larger than 0.3 in most cases. For RECinf the correlations were generally higher than for RECall, thus, the results supported the hypothesis that RECinf is the more reliable trait. However, this result should be considered with caution as fewer cells were investigated and fewer colonies were assessed, as also indicated by the reported standard errors. The repeatability of SMR was very low and, considering the standard errors, not essentially different from zero. The correlation between SMR1 and SMR3, with a very small number of observations, was an exception to this rule. Good repeatability of REC in comparison with SMR might be explained by the fact that REC represents a behavior of the workers directly, while SMR is a more indirect measurement as it is influenced by a variety of causes such as the social hygienic behavior of the workers, properties of the brood, recapping, as well as properties of the mites. The reason for the very low repeatability of SMR might also be an insufficient sample size. In our study, the average sample size was 24 single infested cells per colony. According to a recent study by Mondet et al. [23], the real SMR values then could deviate more than 20% up and down from the observed scores. At least 100 single infested cells would be needed to score SMR with less than 12% deviation up and down. However, an analysis of such large numbers would require a significant amount of time and would not be realistic for field performance tests. Data from 50 to 60 MiniPlus colonies repeatedly tested each year at the institute in Kirchhain revealed repeatability values of 0.35 to 0.70 for RECall, and 0.01 to 0.09 for SMR (Büchler, unpublished data 2019). This seems well in line with the findings of this study, even though these colonies were artificially infested with mites, while in the present study we worked with natural infestation levels. We are not aware of published data on the repeatability of REC or SMR thus far, except a recent publication [24] where the estimated repeatability for SMR was 0.43 ± 0.11 when readings were only 10 days apart, and 0.17 ± 0.09 when they were separated by a longer time and spread over the season. The latter estimate is close to ours, both in terms of timing as in the level of the repeatability. It should be noted that the genetic background of the colonies in this recent study was very diverse (Eynard, S.E., personal communication), which probably increases the level of repeatability as compared to our study and has limited value in the context of performance test and a selection program.

There are heritability estimates in the literature, however, where repeatability reflects the upper limit of heritability. Heritability for SMR varied between 0.06 ± 0.48 and 0.46 ± 0.59 [25] while higher heritability values were found [2,26]. According to Harbo and Harris [27], SMR was closely linked with VSH. Villa et al. [28] measured the change in brood infestation during one week after introducing infested combs either into colonies selected for VSH or into unselected control colonies. They found a much higher repeatability in the group that was selected for VSH. If a similar effect holds for SMR, on top of sample size, the low repeatability in the Croatian dataset may be due to the low average SMR level of 26%. Note that in the other two populations in this experiment the average level of SMR was only slightly higher.

With repeatability values in the order of 0.15 to 0.35 for PIN and REC, the testing methods used in this study identify the pin test and recapping rates as stable properties that can be reproduced within the test season. In general, to increase the usefulness of these traits for performance testing, repeated measures are recommended [11], as these increase the accuracy of the breeding value estimation. In contrast, SMR cannot be reproduced and each reading must be considered as a one-time assessment. This argument for SMR seems supported with high repeatability for short intervals and low repeatability for longer intervals [24].

### 4.2. Correlations between Different Mite Infestation Traits and Design of Performance Test

We discuss the correlations between different mite infestation traits and the design of the performance test in the same paragraph, because the understanding of these correlations and the design of the test share similar arguments.

First of all, the correlation between traits based upon the same reading must be distinguished from traits without common reading. For instance, MPG and BINFa both are calculated from BINF and NMF, and therefore, the high correlation found is expected. The same is true for b3 and b5, which are also highly correlated. Other trait comparisons have a partial overlap, for instance MPG and BINFa with b3 and b5. The correlations found reflect this fact. Secondly, BINF1 … BINF5 and BRINF are absolute parameters describing an infestation, while MPG, BINFa, b3 and b5 are relative parameters indicating an infestation growth.

BRINF is independent from the other infestation traits; thus, the relatively high correlation to MPG (0.35) and BINFa (0.41) is particularly meaningful and indicates a close connection of mites found in brood (an absolute parameter) and infestation growth on bees. We can observe major differences between the countries here—while in Germany the correlations between BRINF and BINFa are very high (0.6), in Austria and Croatia they are low (0.13, 0.16). This must be seen in the context that the average mite infestation levels in Germany were intentionally higher than in Austria and Croatia.

As emphasized by Guichard et al. [6], drifting is one of the main challenges in measuring mite infestation development. Pfeiffer and Crailsheim [29] estimated 13–42% alien bees in neighboring colonies depending on their positions in the apiary and the season. Similar results were reported by Jay [30], who found drifting rates between 11.5–24.7% within 7 days and 24.4–40.5% within 21 days after brood emergence. Therefore, to optimize testing for mite development, much attention must be paid to the design of apiaries where performance testing is carried out. Colony arrangement in squares with the entrances facing in different cardinal directions reduced drifting compared to arranging colonies in rows; moreover, colored entrance boards also had a positive effect [30]. While this should be regarded as good practice in common test apiaries [11], even longer distances (e.g., 70 m) between the hives showed an additional benefit [17]. Such distances might be impossible to realize when at the same time a minimum number of colonies needs to be kept under comparable environmental conditions, as required to separate genetic and environmental effects.

With regard to mite invasion, a clear seasonal pattern with low mite invasion in spring, but high values in summer until autumn was found [19]. A tendency of highly infested workers to enter other colonies was considered, which might result in an equalization or even inverse infestation rate of colonies with different levels of mite resistance.

A significant effect of the infestation level on invasion rate of mites might also explain why we found a negative correlation of BINFa and b3 on the untreated and more highly infested colonies in Germany, while there is a positive correlation measured in the lower infested test populations in Austria and Croatia (Appendix A).

Perhaps in practice the seriousness of drifting might be quantified by the repeatability of BINF. In our study, spanning a period of 6 weeks, drifting might explain the somewhat lower repeatability of BINF in Germany as compared to Croatia and Austria and underlined the importance of repeated measurement.

A longer period of undisturbed infestation development is useful to identify differences between the colonies. With increasing infestation levels, however, and especially in later summer to autumn, an increasing transfer of mites within the apiary has to be taken into account. It might therefore be useful to start bee sampling for mite infestation as soon as a minimum bee infestation is noticed in most of the colonies (e.g., 1% infestation level). If more measurements can be taken, they should better be taken in an early phase, because the later measurements are more disturbed by secondary effects, indicated by the low correlation of BINF4 and BINF5 to the earlier measurements. A larger number of samplings also increase the reliability of a growth factor (comparable to b5 for Austria in our study), which is an alternative to BINFa.

A third infestation parameter, BRINF, requires more effort to measure, but might be less affected by external factors and could be more useful than expected to date. Additional data will be required to establish its usefulness as a new parameter.

### 4.3. Correlations between Behavioral Traits

Correlations between the pairs of related behavioral traits, PINem with PINop and RECall with RECinf, are high, while the correlations between behavioral traits of different types are lower but, in the combined data, essentially positive (Table 7). A significant positive correlation is found between PINop and RECinf which could indicate the underlying behavioral trait is more closely related than for other trait pairs, e.g., PINem to RECall. Indeed, the critical element of the removal of a damaged larva is the recognition indicated by starting to remove the brood cap [31]. Thus, the relatively close connection of the initiation of the brood cap removal in the pin test (PINop) and the selective recapping (RECinf) is not surprising.

Out of measured behavioral traits (PINem, PINop, RECall and RECinf), RECinf is the one with the highest correlation to SMR, meaning that higher rate of recapping mite infested cells increases the proportion of non-reproducing mites (Table 7). Novel studies indeed identified recapping of brood as a key resistance mechanism in several populations [32,33]. Presence of Varroa mites in a brood cell elicits hygienic removal of infested cells by workers [34,35] and REC and VSH, as different expressions of hygienic behavior, are closely linked to each other. However, it is reported that opening the cell has a crucial role as both hygienic and non-hygienic bees are equally capable to recognize and remove dead or diseased brood once opened [36]

In conclusion, the different types of behavioral traits (pin test, recapping, SMR) are all connected with each other, albeit with a relatively low correlation. Thus, the range of different mechanisms of brood hygiene that honeybee colonies express may contribute more or less independently to the overall resistance of colonies. A similar connection is found between removal of Varroa infested brood (VSH) and hygienic removal of dead brood either freeze-killed [37] or pin killed [22].

### 4.4. Relationships between Behavioral Parameters and Mite Infestation

With the exception of RECall, the behavioral traits are negatively correlated with the mite infestation traits as expected based on the hypothesis that stronger hygienic behavior and suppressed mite reproduction reduces mite population growth, and subsequently, infestation. The correlations were only between −0.1 and −0.2; however, they were significantly different from zero. The highest negative correlations for all countries combined were found for SMR and RECinf with BRINF, and PINop with BINFa. The results differed considerably between the countries, however. For instance, the high negative correlations between BRINF and both RECinf and SMR in the Austrian dataset was not found in the Croatian dataset, where it is not significantly different from zero, while in the German dataset it is significantly negative, but with a much lower correlation. Additionally, in the Croatian dataset, there is a highly negative correlation between PINop and BRINF (−0.37), which was not found in the German dataset. Our findings indicate that these correlations are based on causal connections depending on specific conditions such as the average infestation levels and the timing of sample collection.

This may also explain inhomogeneous findings on the relevance of hygiene behavior for mite infestation development in the published literature. Negative correlations of PINem and PINop with infestation parameters in this research suggest a negative impact of hygienic behavior on mites, especially on the newly introduced mite population growth parameter BINFa. While Ibrahim et al. [38] found significant correlations between hygiene behavior and BRINF and, to a lower extent, BINF, several other studies did not observer such correlations [39,40,41,42]. Comparing the correlations of PINem and PINop to the infestation traits in our study, it can be concluded that PINop is the more promising behavioral trait to predict suppressed mite population growth. However, up to now, all other studies used the proportion of totally removed cells to estimate the hygiene behavior, either for the pin or frozen brood assay.

It seems that hygienic behavior is not a good indicator of possible resistance traits in unselected populations [43]. Although highly hygienic colonies may slow down mite population growth significantly [44], the proportion of such colonies in average populations seems to be small. Difficulties in associating hygienic behavior with mite infestation may also arise from the fact that bees selectively remove brood infested with mites carrying DWV, while mites with low viral loads could be neglected [45].

Recent experiments recognized recapping of brood cells as a key mechanism of resistance in surviving populations [32,33]. Substantial correlations of REC with BINF and BRINF were found by Villegas and Villa [46]. It is important to state from our study that recapping of infested cells (RECinf) is obviously the more relevant trait, as it is independent of the infestation level, while recapping of all cells (RECall) is not. Even more, the significant positive correlation between RECall and BRINF in Croatia, and BRINF and BINFa in Germany indicates that indiscriminate opening of brood cells may be contra productive such that resistance of colonies depends on a highly specific identification and recapping of Varroa-infested cells.

We calculated phenotypic correlations of breeding colonies registered in BeeBreed for which mite infestation and mite fall as well as PIN was measured. The correlation between PIN and BINFa was −0.06 ± 0.01, and adjusted for the effect of season × apiary, it was −0.08 ± 0.01. The correlation between PIN and MPG was −0.02 ± 0.01 and −0.07 ± 0.01, respectively. Thus, the correlations are similarly low, while being significantly different from zero due to the large number of observations.

A suppression of mite reproduction (SMR) is often seen as a crucial indicator of mite resistance [2,27], and several studies reported negative correlations between SMR and mite infestation [26,47,48,49]. However, in an analysis of SMR in 13 European countries on nine different genotypes, correlations between SMR and brood infestation were found to be not significantly different from zero [23]. Similarly, no significant correlations between SMR and mite infestation after two generations of bi-directional selection for mite population growth were found [16]. Harris et al. [15] described that in non-resistant stocks, behavioral traits might explain just a small part of the mite infestation variability. This might contribute to the low correlation coefficients in our study. With 26.3% in Croatia to 32.8% in Germany, the average levels of SMR are much lower than those reported for resistant populations in Gotland (Sweden) or Avignon (France) [50].

### 4.5. Breeding Objective

The performance test of a honeybee colony establishes the phenotype with respect to traits that represent the overall breeding objective that includes honey yield, gentleness, calmness, low swarming drive, disease resistance and overwintering strength. In this discussion, we focus on the Varroa resistance, for which mite infestation is included in the breeding objective.

Varroa resistance has several aspects, including long-term survival under Varroa infestation pressure, honey yield and absence of other diseases even with Varroa infestation, as well as sustainable reduction of Varroa mite population. However, the scope of these aspects is too complex to capture and simpler breeding objectives are needed to represent Varroa resistance as well as possible. Reduction of mite population is clearly the most accessible of them. Parameters such as BINFa, MPG, b3, b5, BRINF or MFOA represent this breeding objective, while behavioral parameters such as SMR, REC and PIN may contribute as indirect selection parameters.

Concerning the mite infestation traits BINFa and MPG, our results suggested that BINFa should be preferred over MPG because of higher repeatability and higher correlations to behavioral traits. MFOA was included in our study as it might become an interesting parameter once beekeepers stop winter treatments and instead apply efficient mite reduction by summer brood interruption [51]. Finally, decisions on this issue strongly depend upon levels of heritability and genetic correlations but also upon the ease of measurement, as a balance needs to be found as to which combination of measurements best captures Varroa resistance, given limited resources such as time and money.

### 4.6. Genetic Parameters and Response to Indirect Selection

Repeatability of the measured parameters and consistent correlations among different parameters show a general suitability for selection. However, for a strategy of sustained selection progress, heritability, genetic correlations and reliable models for breeding values have to be determined. For a trait with high heritability, it is relatively easy to achieve a selection progress, while for a trait with lower heritability a larger population and more consistent testing is needed. The precision of a breeding value model increases when both worker and queen effects are considered, which are mostly negatively correlated with each other. The higher the negative correlation, the more important it is to select on both effects.

For pin test, several estimations of heritability have been reported, most recently by Hoppe et al. (2020, submitted) as 0.52. For SMR, the reported heritability was up to 0.46 [25]. Thus, the heritability of behavioral traits can be very high. For mite population growth, low heritability has been reported, e.g., 0.05 (Hoppe et al., 2020, submitted). Thus, in honeybee breeding for Varroa resistance we face the choice from parameters that are easy to measure, but provide a low contribution to the breeding objective (because of low heritability or low genetic correlation with objective traits), and traits that are tedious to measure, but contribute more (because of high heritability or high genetic correlation). As an illustration of the relevance of heritability estimates and genetic correlations, we discuss the value of indirect selection for a breeding-objective trait, selecting for another trait, in that way getting an impression of its possible contribution. For this, genetic parameters are essential. The response of a trait y selection of the trait x  can be written as [52]:(7)$$\frac{CR_{y}}{R_{x}}=r_{A}\frac{i_{y}}{i_{x}}\frac{h_{y}}{h_{x}}$$
where $CR_{y}$ is the correlated response to selection in x when selecting for y, and $R_{x}$ is the response to selection in x when selecting for x  itself. Furthermore, $r_{A}$ stands for the additive genetic correlation between x  and y, i for intensity of selection and h for the square root of heritability. In our case, x included BINFa and b3, which can be measured practically on very many colonies, while traits such as SMR and REC are usually measured on fewer colonies such that their intensity of selection is substantially smaller than for BINFa and b3. For PIN, however, we can assume that both intensities of selection are equal such that we only need estimates of the heritability values for BINFa, b3 and PIN, and estimates of $r_{A}$. Recently, Hoppe et al. (2020, submitted) estimated the genetic parameters for the combination of worker and queen effect [53] for the main Carnica population within BeeBreed as $h_{BINFa}^{2}=0.05$ and $h_{PIN}^{2}=0.52$, while $r_{A}=-0.48$. The sign is in the expected and desired direction: the higher the PINem, the lower the BINFa. Taking these values $\frac{CR_{PIN}}{R_{BINFa}}=-1.62$, such that indirect selection for PINem is more effective than direct selection for BINFa. To complete an analysis like this, the issue actually is not selecting for either BINFa or PINem, but selecting for both. The repeatability for BINF of 0.55 in our German dataset and the heritability of 0.05 suggests that most part of the repeatability is due to permanent environmental effect and not due to additive genetic effect. Indirect selection for BINFa through SMR only will be beneficial when SMR has a substantial heritability and a sufficiently high genetic correlation. The very low repeatability we found for SMR does not exclude a substantial heritability, but it would imply that the genetic correlation between repeated measurements is close to zero, in line with the idea of a one-time measurement. Additionally, note that Equation (7) holds for selection on single phenotypes, while in practice, information on relatives is used, and the accuracy of selection is no longer $\sqrt{h^{2}}$, but larger, and the accuracy of traits are more similar in size than their heritability. These exercises are useful, however, to decide in which activity to invest given limited resources.

These findings underline the importance of estimating genetic parameters when considering the value of traits for selection. For SMR and REC as yet there are no reliable estimates of genetic parameters. Extensive selection work of Arista Bee Research (aristabeeresearch.org) with several breeds and a project funded by the German Bundesanstalt für Ernährung und Landwirtschaft (https://service.ble.de/ptdb/index2.php?detail_id=2103579&site_key=293&stichw=SMR&zeilenzahl_zaehler=2#newContent) may provide the necessary information to judge the value of these traits for selection in the near future.

It should be emphasized that it is important to study genetic parameters for specific populations even though this is a serious problem for small datasets. Perhaps the large Carnica main population in BeeBreed (with 10,000 records added annually) may serve as a reference population and it may be possible to judge whether the genetic parameters in a specific population differ significantly from kind of a consensus value from such a reference population. As an example, it proved not possible to detect additive genetic variance for MPG in Swiss Carnica and Mellifera in datasets of about 1000 records each [54]. The issue is whether this means that its heritability is zero or that, considering the fairly small dataset, a consensus value would be better to use for practical purposes.

## 5. Conclusions

Measurements of infestation (BINF), brood removal (PINop) and recapping (REC) are sufficiently repeatable (0.55, 0.33 and 0.35, respectively), which qualify them as suitable selection traits. Very low repeatability of SMR suggests that the SMR protocol has to be improved, e.g., to define the appropriate time point in the season.

Repeatability of bee infestation drops in late season. Thus, we suggest either restricting the time of measurements in the season or taking special care to avoid drifting in the design of the performance test.

To select for mite infestation we suggest three traits, mite infestation in summer adjusted for initial mite infestation in spring by regression, exponential mite population growth in summer and brood infestation. Together these traits can give a reliable picture of resistance to mite population development. Repeated measurement of mite infestation is strongly recommended as it enables a more accurate estimate of mite population growth.

The proportion of open cells in the pin test and the rate of recapping of infested cells are phenotypically connected traits and significantly correlate with mite infestation. They are suitable selection criteria for mite resistance, although the ultimate choice of selection traits primarily depends upon genetic parameters (heritability and genetic correlations for worker and queen effect) and not phenotypic ones (repeatability and phenotypic correlations), as presented in our paper. We suggested to arrive at a kind of reference parameters estimated in large datasets and to judge whether estimates in small populations should be taken to either or not deviate from those.

Although differences in mite reproduction (SMR) are negatively correlated with brood infestation (BRINF) and mite population increase (b3), a repeatability of measurements close to zero indicates that its application might not be efficient for selective breeding programs.

## Figures and Tables

**Figure 1 insects-11-00618-f001:**
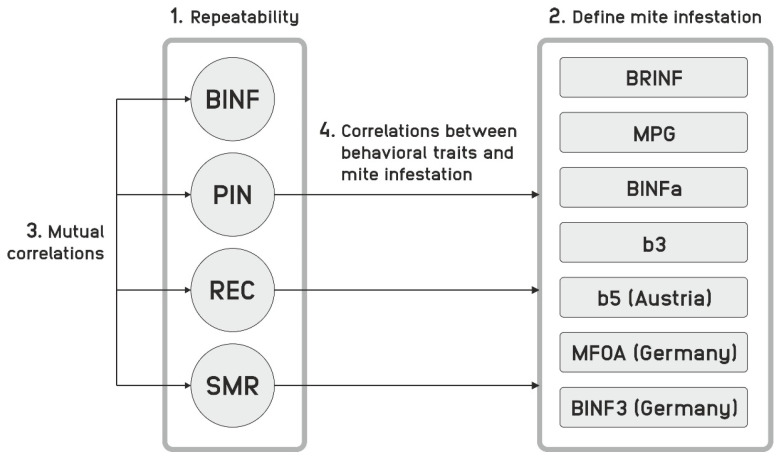
The four paragraphs in the results section.

**Table 1 insects-11-00618-t001:** Overview of all measured or derived traits (see description in text for details).

Abbreviation	Parameter
NMF	Natural mite fall [mites/day]
BINF *	Bee infestation [mites /10 g of bees]
MFOA	Mite fall after oxalic acid treatment of broodless colonies in July
BRINF	Brood infestation rate [%]
MPG *	Mite population growth
BINFa	Bee infestation adjusted for NMF
b3	Growth factor from BINF1 to BINF3
b5	Growth factor from BINF1 to BINF5
PIN	PIN test for hygiene behavior
PINop *	Cells opened [%]
PINem *	Cells completely emptied [%]
REC	Recapping of brood cells
RECall *	Recapping of all inspected cells [%]
RECinf *	Recapping of all infested cells [%]
SMR	Cells without mite reproduction [%]

*: the term can be followed by a number, indicating the sequence of repeated samples.

**Table 2 insects-11-00618-t002:** Numbers of observations, means and standard deviations (Stdev) for BINF, BRINF, NMF, PINem, PINop, RECall, RECinf and SMR (see Table 1 for description and units of traits).

	Austria			Croatia			Germany		
	N	Mean	Stdev	N	Mean	Stdev	N	Mean	Stdev
**BINF ^1^**	147	0.91	1.14	114	1.33	2.27	207	2.76	2.07
**BRINF**	147	26.2	17.5	135	11.0	12.9	207	23.7	16.3
**NMF**	147	0.19	0.24	135	0.34	0.94	207	2.06	1.51
**PINem ^2^**	134	38.4	14.1	126	48.6	26.8	207	37.4	16.5
**PINop ^2^**	147	85.9	15.2	53	71.8	23.5	207	83.5	17.0
**RECall**	147	37.0	28.0	135	23.3	21.8	207	24.4	25.7
**RECinf**	147	59.4	31.6	135	50.6	32.1	206	43.0	38.4
**SMR**	147	30.2	15.4	133	26.3	13.5	206	32.8	15.4

^1^ Average of three measurements, BINF1, BINF2 and BINF3. In the German dataset, BINF3 was divided by 4.31, see text. ^2^ Average of two measurements.

**Table 3 insects-11-00618-t003:** Correlations between repeated measurements of BINF (1–3), above the diagonal and standard errors below the diagonal.

	Austria			Croatia			Germany			Combined		
	1	2	3	1	2	3	1	2	3	1	2	3
**1**		0.68	0.49		0.85	0.76		0.56	0.51		0.58	0.52
**2**	0.05		0.57	0.04		0.64	0.05		0.67	0.03		0.66
**3**	0.06	0.06		0.06	0.09		0.05	0.04		0.04	0.03	

**Table 4 insects-11-00618-t004:** Correlations between repeated measurements of PINem (PINem1-2) and PINop (PINop1-2 for Austria and Germany), above the diagonals and standard errors (below diagonals).

	Austria		Croatia		Germany		Combined	
	1	2	1	2	1	2	1	2
**PINem1**		0.31		0.24		0.15		0.20
**PINem2**	0.11		0.13		0.07		0.05	
**PINop1**		0.41				0.32		0.34
**PINop2**	0.10				0.06		0.05	

**Table 5 insects-11-00618-t005:** Correlations between repeated measurements of RECall and RECinf (above diagonals) and their standard errors (below diagonals). The number of observations ranges between 4–37. For the correlations in italics, the numbers of observations are smaller than 20.

	RECall						RECinf					
	1	2	3	4	5	6	1	2	3	4	5	6
**1**		0.44	0.27	0.36	0.14	0.40		0.95	0.78	0.74	0.99	
**2**	0.12		0.35	0.12	0.34	0.21	0.07		0.75	0.30	*0.82*	*0.47*
**3**	0.14	0.13		0.41	0.48	0.18	0.28	0.20		0.22	0.28	0.33
**4**	0.13	0.15	0.13		0.23	0.15	0.32	0.41	0.22		0.07	0.06
**5**	0.14	0.13	0.12	0.14		0.36	0.01	0.14	0.21	0.18		0.26
**6**	0.13	0.15	0.16	0.16	0.14			0.45	0.22	0.19	0.16	

**Table 6 insects-11-00618-t006:** Correlations between repeated measurements of SMR (above Diagonal) and their standard errors (below diagonal). The number of observations ranges between 4–37. For the correlations in italics, the numbers of observations are smaller than 20.

	1	2	3	4	5	6
**1**		−0.03	0.93	0.13	−0.10	
**2**	0.71		0.21	0.02	0.06	−0.11
**3**	0.09	0.43		−0.01	0.07	−0.19
**4**	0.70	0.45	0.23		0.15	0.45
**5**	0.70	0.45	0.23	0.18		−0.04
**6**		0.57	0.23	0.15	0.17	

**Table 7 insects-11-00618-t007:** Correlations between different definitions of mite infestation (above the diagonals) and their standard errors (below the diagonals).

	**Austria**							**Croatia**				
	**BRINF**	**MPG**	**BINFa**	**b3**	**NMF**	**b5**		**BRINF**	**MPG**	**BINFa**	**b3**	**NMF**
**BRINF**		0.17	0.12	0.13	−0.08	0.20			−0.11	0.16	−0.11	0.62
**MPG**	0.08		0.84	0.29	−0.40	0.01		0.14		0.79	0.56	-0.35
**BINFa**	0.08	0.02		0.36	0.01	0.15		0.14	0.05		0.32	0.00
**b3**	0.08	0.08	0.07		0.06	0.70		0.14	0.10	0.13		−0.13
**NMF**	0.08	0.07	0.08	0.08		0.13		0.09	0.12	0.14	0.14	
**b5**	0.08	0.08	0.08	0.04	0.08							
	**Germany**							**Combined**				
	**BRINF**	**MPG**	**BINFa**	**b3**	**NMF**	**MFOA**	**BINF3**	**BRINF**	**MPG**	**BINFa**	**b3**	**NMF**
**BRINF**		0.50	0.60	−0.12	0.02	0.41	0.55		0.35	0.41	−0.01	0.04
**MPG**	0.05		0.73	−0.10	−0.57	0.26	0.57	0.04		0.75	0.10	−0.50
**BINFa**	0.04	0.03		−0.16	0.01	0.31	0.90	0.04	0.02		0.02	0.01
**b3**	0.07	0.07	0.07		−0.23	−0.09	0.08	0.05	0.05	0.05		−0.16
**NMF**	0.07	0.05	0.07	0.07		0.07	0.11	0.05	0.04	0.05	0.05	
**MFOA**	0.06	0.07	0.07	0.07	0.07		0.36					
**BINF3**	0.05	0.05	0.01	0.07	0.07	0.06						

**Table 8 insects-11-00618-t008:** Correlations between behavioral traits (above the diagonals) and their standard errors (below the diagonals).

	**Austria**					**Croatia**				
	**PINem**	**PINop**	**RECall**	**RECinf**	**SMR**	**PINem**	**PINop**	**RECall**	**RECinf**	**SMR**
**PINem**		0.51	0.24	0.24	0.09		0.46	−0.12	−0.06	0.04
**PINop**	0.06		0.27	0.35	0.24	0.11		0.06	0.22	0.09
**RECall**	0.08	0.08		0.84	0.36	0.09	0.14		0.75	0.02
**RECinf**	0.08	0.07	0.02		0.42	0.09	0.13	0.04		0.11
**SMR**	0.09	0.08	0.07	0.07		0.09	0.14	0.09	0.09	
	**Germany**					**Combined**				
	**PINem**	**PINop**	**RECall**	**RECinf**	**SMR**	**PINem**	**PINop**	**RECall**	**RECinf**	**SMR**
**PINem**		0.43	0.21	0.25	0.08		0.44	0.13	0.16	0.07
**PINop**	0.06		0.23	0.25	−0.08	0.04		0.20	0.26	0.04
**RECall**	0.07	0.07		0.84	0.00	0.05	0.05		0.82	0.11
**RECinf**	0.07	0.07	0.02		0.00	0.05	0.05	0.01		0.14
**SMR**	0.07	0.07	0.07	0.07		0.05	0.05	0.04	0.04	

**Table 9 insects-11-00618-t009:** Correlations between behavioral traits and three traits describing mite infestation. For the Austria dataset the standard errors are 0.08, for the Croatian dataset they vary between 0.08 and 0.15, for the German dataset they are 0.07 and for the combined dataset 0.05 (correlations significantly different from zero (*p* < 0.10 two-sided test) in bold letters).

	Austria			Croatia			Germany			Combined		
	BRINF	BINFa	b3	BRINF	BINFa	b3	BRINF	BINFa	b3	BRINF	BINFa	b3
**PINem**	−0.06	−0.12	−0.03	−0.13	0.09	−0.01	**−0.13**	−0.09	0.01	**−0.10**	−0.06	−0.01
**PINop**	**−0.23**	**−0.21**	−0.03	**−0.37**			−0.02	**−0.18**	0.08	**−0.16**	**−0.19**	0.03
**RECall**	−0.09	−0.02	−0.08	**0.22**	−0.13	−0.01	0.10	**0.13**	**−0.13**	0.06	0.07	**−0.10**
**RECinf**	**−0.37**	**−0.15**	**−0.18**	0.02	0.01	0.11	−0.11	−0.04	**−0.11**	**−0.17**	−0.05	**−0.10**
**SMR**	**−0.35**	−0.07	−0.08	−0.05	0.03	−0.12	−0.09	−0.05	**−0.14**	**−0.17**	−0.04	**−0.12**

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
