# Peer review of "Evaluation of Traits for the Selection of Apis Mellifera for Resistance against Varroa Destructor"

_insects, 2020, doi:10.3390/insects11090618_

Round 1

Reviewer 1 Report

General Comments

The paper adds some interesting knowledge in the operational strategies that may help to select honeybees for Varroa resistance based on a mite infestation levels and specific behavioural traits. The authors estimate repetabilities and phenotypic correlations across different mite infestation traits, hygienic behaviour (pin test), brood recapping (REC) and  suppressed mite reproduction (SMR) in three datasets of Apis mellifera carnica from three countries: Austria (147 records), Croatia (135) and Germany (207).

However the paper suffers from a presentation of both the material and methods and the results that is not enough direct and synthetic, including some vague or incorrect words in some sentences (see specific comments for further explanations). In addition, some other calculations done by the authors (based on beebreed data) are only presented in the discussion part. It should either be included in the previous sections or completely removed from the paper. Therefore an important rewriting is necessary in order to better present the work.   

In addition, I think it is important that the authors present in a clearer manner what they consider as breeding goal(s) and what are the associated selection criteria. In my opinion, the breeding objective is what the autors named the ultimate goal or the golden standard, i.e a high survival of colonies without curative treatments (lines 44-45). All other traits should be named all throughout the paper « selection criteria » as they are only indirect measures that may used to select for improving the survival of colonies. In line 69, it seems that the authors finally consider that mite infestation is the operational breeding goal and that selection criteria are behavioural traits. If so, it should be clearly stated and kept consistent throughout the paper. If not, please try to clarify better throughout the paper when you are considering it as a direct selection criterion (corresponding to a  breeding goal) or an indirect one.

 Specific comments 

Table 1. The last column is unnecessary and should be removed for the sake of clarity. 

Lines 121 to 129. The BINF measurements seems to be very different from one country to another. Therefore we may wonder whether it is relevant or not to combined data in a fina analysis. In addition this point should be discussed in the discussion section as BINF appears to be the more relevant mite infestation trait to promote for selection. Recommendations on the way to record it should be given. 

Line 134 to 136. It does not appear very useful for the paper to present results for MFOA that was only recorded in Germany. For the sake of clarity in presenting methods and results, I suggest that the authors remove this trait from the paper…If they are willing to maintain it, its usefulness should be clarify in the discussion section. L

ine 163-164. It should be accurately defined the thresholds (for each country) of brood sample size and infestation under which only 10 cells were analyzed rather than the usual 35. In addition, because the infestation rates are not really high in that study, it may be think that this case occurs quite frequently and therefore the SMR values are not at all accurate. The proportion of data with the 35 cells recorded and the proportion of data with only 10 cells recorded should be given for each country. This may at least partially explained the low repeatability for SMR values. 

Table 2. Units should be given for each trait Line 176. What means “interestingly”? Please explain without being subjective 

Line 197. Justify the division by 4.31 with further detail or references. 

Lines 199-207. The usefulness of b3 and b5 is not clear for me in comparison to the BINFi. Please explain the potential usefulness of these traits. 

Table 3 should be put in supplementary table for the sake of clarity of the paper. 

Line 221. Remove the first sentence. 

Line 224. Please be explicit to say that the traits analyzed are what you call “deviations”, ie your initial traits corrected for the significant fixed effects of apiary, season…. 

Lines 227-228. Change BINF1 into BINFi in the text and in the equation. 

Lines 229-230. We may wonder if rather than to divide by a “magic” 4.31 BINF3 for the German dataset considering an average BINFa only for BINF1 and BINF2 measures would have led to more comparable results for the 3 countries. 

Line 231. “Deviations” is not really clear here, I suggest to used rather the terms corrected or adjusted traits

Lines 236-238. It is not necessary to give confidence intervals when se are given. In addition if you are willing to give them, keep the sentences for a legend of the supplementary table II and remove the lines 236-238 for the main text.

Lines 240-253. The beginning of the results section is very surprising. In my opinion, Figure 1 is not at all helpful and should be removed. In addition, the paragraph should be shortened to introduce the 4 points of the results section.

Tables 4 to 8 should be improved to be more easy to read. Maybe a good option would be to consider only the information that is similar for all countries and to add a supplementary table for the data that are only available in a specific country.In addition, it would be good to have in brackets the number of records to derive each correlation rather than comments like “With correlations in italics the numbers of observations are smaller than 20 » in table 7.

 Line 332-33. Reformulate more explicitely Line348. 17 apiaries ? This number is not consistent with lines 80-108 in the material section

Line 349. Reformulate because the differences were not reduced to …but corrected for

Line 361 and in the next parts of the discussion. Results for Beebreed are given : they should be presented earlier if the results were obtained for the purposes of this study or references to previous work should be given .

Lines 377-378. I do not understand the sentence. What is a « technical variance » ?

Line 383. Explain what is AGT Lines 384-393. Reformulate all the paragraph because everything is mix up, between different measures, different populations and same measurement over time in the same population.

 Line 404-405. It is the point that I noticed in the method section. I think it is a major drawback of your study for SMR and it should clearly modulate your conclusions about SMR. In addition if the average number of cells was only 24…what was the median number…probably less than 20 …ie half the number recommended for the measure !

Lines 438-440. Reformulate Lines 481-484. I do not understand what points in your study is justyfing your statements  

Lines 560 to 637. The end of the discussion section is very important and should deserve a certain amount of rewriting with clear breeding objectives, direct and indirect selection criteria definitions. 

Start the conclusion with line 641. 

Lines 646. Drifting should have been defined when the term was used for the first time in the discussion section.

Reviewer 2 Report

Infestation with Varroa destructor is a serious cause of bee colony (Apis mellifera) losses on a global level. The authors based their analyses on datasets of Apis mellifera carnica from three countries: Austria (147 records), Croatia (135) and Germany (207). The results showed that bee infestation in summer, adjusted for the level of natural mite fall in spring, is a suitable breeding objective, and suggested adding brood infestation rate and the increase rate of bee infestation in summer. Repeatability for bee infestation rate was about 0.55, for cells opened in pin test about 0.33, for recapping 0.35 and for SMR virtually zero. Their research gives good advice for selection for mite infestation in bee breeding. I suggest some revision of the text, mainly of some parts of discussion.

Minor comments:

1. Line 386-387, need reference here for this data.

2. Line 409-411, it is better not to include data unpublished.

Round 2

Reviewer 1 Report

The clarity of the paper has been greatly improved. The authors take in consideration most of my initial comments and give clear explanations why they did not for some of them. Thank you for that. I have no further comment to add.